# Human forebrain organoid-based multi-omics analyses of PCCB as a schizophrenia associated gene linked to GABAergic pathways

Wendiao Zhang [1,2,3,4], Ming Zhang[1], Zhenhong Xu[2,3,4], Hongye Yan[2,3,4], Huimin Wang[2,3,4], Jiamei Jiang[2,3,4], Juan Wan[2,3,4], Beisha Tang [2,3,4,5,6], Chunyu Liu [1,7] ✉, Chao Chen [1,6,8,9] ✉ & Qingtuan Meng [2,3,4,10] ✉

Identifying genes whose expression is associated with schizophrenia (SCZ) risk by transcriptome-wide association studies (TWAS) facilitates downstream experimental studies. Here, we integrated multiple published datasets of TWAS, gene coexpression, and differential gene expression analysis to prioritize SCZ candidate genes for functional study. Convergent evidence prioritized Propionyl-CoA Carboxylase Subunit Beta (*PCCB*), a nuclear-encoded mitochondrial gene, as an SCZ risk gene. However, the *PCCB*'s contribution to SCZ risk has not been investigated before. Using dual luciferase reporter assay, we identified that SCZ-associated SNPs rs6791142 and rs35874192, two eQTL SNPs for *PCCB*, showed differential allelic effects on transcriptional activities. *PCCB* knockdown in human forebrain organoids (hFOs) followed by RNA sequencing analysis revealed dysregulation of genes enriched with multiple neuronal functions including gamma-aminobutyric acid (GABA)-ergic synapse. The metabolomic and mitochondrial function analyses confirmed the decreased GABA levels resulted from inhibited tricarboxylic acid cycle in *PCCB* knockdown hFOs. Multielectrode array recording analysis showed that *PCCB* knockdown in hFOs resulted into SCZ-related phenotypes including hyper-neuroactivities and decreased synchronization of neural network. In summary, this study utilized hFOs-based multi-omics analyses and revealed that *PCCB* downregulation may contribute to SCZ risk through regulating GABAergic pathways, highlighting the mitochondrial function in SCZ.

Schizophrenia (SCZ) is a complex polygenic psychiatric disorder with risk contributed by environmental and genetic factors[1]. Genetic studies such as genome-wide association studies (GWAS) have identified hundreds of common single nucleotide polymorphisms (SNPs) associated with SCZ[2,3]. Most of the SCZ-associated SNPs are noncoding variants located in regulatory DNA elements[4–6], suggesting that gene expression mediates the connection between genetic variants and SCZ phenotypes[7]. Identifying genes whose expression is associated with SCZ phenotypes facilitates discovering SCZ risk genes for downstream functional studies.

By integrating SCZ GWAS and brain expression quantitative trait loci (eQTL) data, several approaches, which are collectively described

as transcriptome-wide association studies (TWAS) have been used to identify SCZ risk genes. These TWAS approaches, including FUSION[8–10], PrediXcan[11], summary-data-based Mendelian randomization (SMR)[2,10,12,13], and joint-tissue imputation approach with Mendelian randomization (MR-JTI)[14], aimed to identify the association between predicted gene expression and SCZ risk. Though MR-JTI could improve gene expression prediction performance in TWAS and provide a causal inference framework[15], experimental validation is still needed.

Here we integrated results from MR-JTI[14] and other published SCZ TWAS datasets[2,8–13] to prioritize promising SCZ risk genes for functional study. Through this integrative analysis, we identified Propionyl-CoA Carboxylase Subunit Beta (*PCCB*), a protein-coding gene that plays important roles in mitochondrial metabolism[16,17], as an SCZ risk gene with the most supporting evidence in our analysis. However, the *PCCB*'s contribution to SCZ risk has not been investigated before. Using human forebrain organoids (hFOs), three-dimensional cell cultures that recapitulate key aspects of the human brain[18], we found that *PCCB* knockdown in hFOs resulted into SCZ pathology-related cellular phenotypes. We also identified that SCZ-associated common SNPs rs6791142 and rs35874192 may regulate *PCCB* expression, supporting that *PCCB* expression may mediate the genetic effects on SCZ risk.

## Results

### *PCCB* is prioritized as a promising SCZ risk gene

To obtain reliable SCZ risk genes for downstream functional study, we integrated multiple published TWAS datasets (Supplementary Data 1) to prioritize genes with sufficient supporting evidence. We also checked whether the prioritized genes are located in SCZ risk-associated gene coexpression module or dysregulated in post-mortem SCZ brains. These analyses prioritized *PCCB*, *GATAD2A*, and *GNL3* as the top three SCZ risk genes (Table 1). Notably, *PCCB* was also identified as an SCZ risk gene in the gene-based MAGMA analysis[19]. Moreover, *PCCB* was located in the gene coexpression module (M2) downregulated in SCZ based on the PsychENCODE data[10]. *PCCB* was also found to be nominally downregulated in postmortem SCZ brains ($P = 0.01$, $FDR = 0.14$) by checking the SZDB database[20], which integrated transcriptome data from the CommonMind consortium[21]. These lines of evidence suggested that *PCCB* expression mediated the genetic effects on SCZ risk. Therefore, we focused on studying how *PCCB* contributes to SCZ risk in this study.

### *PCCB* eQTL SNPs rs6791142 and rs35874192 affect transcriptional activities

Since *PCCB* expression is genetically associated with SCZ, we investigated the functional impacts of SCZ-associated SNPs on *PCCB* expression. Based on the TWAS results used in this study, we retrieved the top SNPs (rs7432375, rs7427564, rs527888, rs66691851) and their linkage disequilibrium (LD) SNPs that were associated with *PCCB* expression. To narrow down to the putatively causal variants, we focused on those eQTL SNPs (eSNPs) that are likely to affect *PCCB* expression in the brain. Since opening chromatin facilitates gene expression activation, we used brain ATAC-seq data from the PsychENCODE consortium[22] to identify *PCCB* eSNPs located in active transcription regions. By integrating PsychENCODE ATAC-seq data and SNP annotation information from the Roadmap Epigenetics Consortium[23], we prioritized six eSNPs (rs6791142, rs35874192, rs900818, rs7616204, rs570621, and rs7349597) (Table 2) that were located in genomic regions strongly suggested as enhancers or promoters in the human brain tissues or neural cell cultures.

We then performed dual luciferase reporter assay (DLRA) in both human neural progenitor cells (hNPCs) and SH-SY5Y cell lines to validate the regulatory effects of the six eSNP-containing DNA elements. For each eSNP, 50 base pairs (bp) eSNP-containing DNA fragment was synthesized and cloned into the upstream of the *PCCB* promoter in the PGL3-basic luciferase reporter vector (Supplementary Fig. S1). In DLRA,

**Table 1 | Prioritized top three SCZ risk genes**

| Gene | MR-JTI | TWAS | | | | SMR | | | Differential expression | Coexpression module |
|---|---|---|---|---|---|---|---|---|---|---|
| | FUSION Wu et al.[14] | FUSION Hall et al.[9] | PrediXcan Huckins et al.[11] | FUSION Gusev et al.[8] | FUSION Gandal et al.[10] | SMR Yang et al.[13] | SMR Gandal et al.[10] | SMR Li et al.[12] | Goncalves et al.[19] | Gandal et al.[10] |
| **PCCB** | $P_{Bonferroni} = 3.17\mathrm{E}{-}11$ | $P_{TWAS} = 5.39\mathrm{E}{-}12$ | $P = 2.05\mathrm{E}{-}08$ | $P_{TWAS} = 3.07\mathrm{E}{-}10$ | $P_{Bonferroni} = 2.42\mathrm{E}{-}05$ | – | $P_{SMR} = 3.74\mathrm{E}{-}10$ | $P_{SMR} = 4.17\mathrm{E}{-}15$ | ↓ $P = 0.01$ | ↓ FDR = 6.71E-03 |
| **GATAD2A** | $P_{Bonferroni} = 1.87\mathrm{E}{-}08$ | $P_{TWAS} = 8.67\mathrm{E}{-}11$ | $P = 2.18\mathrm{E}{-}10$ | $P_{TWAS} = 8.83\mathrm{E}{-}07$ | $P_{Bonferroni} = 6.98\mathrm{E}{-}09$ | $P_{SMR\text{-}multi} < 1.00\mathrm{E}{-}05$ | $P_{SMR} = 2.21\mathrm{E}{-}10$ | – | – | ↑ FDR = 5.03E-03 |
| **GNL3** | $P_{Bonferroni} = 1.58\mathrm{E}{-}18$ | – | $P = 1.39\mathrm{E}{-}11$ | $P_{TWAS} = 6.00\mathrm{E}{-}07$ | $P_{Bonferroni} = 8.24\mathrm{E}{-}03$ | $P_{SMR\text{-}multi} < 1.00\mathrm{E}{-}05$ | $P_{SMR} = 4.71\mathrm{E}{-}09$ | $P_{SMR} = 4.53\mathrm{E}{-}13$ | – | – |

↓, Downregulation in SCZ; ↑, Upregulation in SCZ. P values are adjusted by the Benjamini-Hochberg or Bonferroni methods.

**Table 2 | Prioritized six eQTL SNPs for *PCCB***

| Source | References | Best GWAS SNP | LD eSNP | LD (r²) | Ref/Alt | Chromatin states (Core 15-state model) | PsychENCODE ATAC-seq Peak |
|---|---|---|---|---|---|---|---|
| MR-JTI | Wu et al., Mol. Neurobiol.[14] | – | – | – | – | – | – |
| TWAS | Hall et al., Hum. Mol. Genet.[9] | rs7432375 | rs6791142 | 0.81 | T/C | 7_Enh | YES |
| | | | rs35874192 | 0.62 | G/C | 1_TssA | YES |
| | | | rs900818 | 0.63 | C/T | 1_TssA, 2_TssAFlnk | YES |
| | Huckins et al., Nat. Genet.[11] | – | – | – | – | – | – |
| | Gusev et al., Nat. Genet.[8] | rs7432375 | rs7616204 | 0.64 | C/T | 7_Enh | YES |
| | Gandal et al., Science[10] | rs7427564 | rs6791142 | 0.73 | T/C | 7_Enh | YES |
| SMR | Gandal et al., Science[10] | rs527888 | rs570621 | 0.95 | A/G | 7_Enh | YES |
| | | | rs7349597 | 0.69 | C/T | 1_TssA, 2_TssAFlnk, 7_Enh | YES |
| | Li et al., Nat. Genet.[12] | rs66691851 | rs7349597 | 0.6 | C/T | 1_TssA, 2_TssAFlnk, 7_Enh | YES |
| | | | rs570621 | 0.81 | A/G | 7_Enh | YES |
| | | | rs6791142 | 0.6 | T/C | 7_Enh | YES |

1_TssA, Active transcription start site (TSS); 2_TssAFlnk, Flanking active TSS; 7_Enh, Enhancer; Ref/Alt, Reference /Alternative allele.

the eSNPs rs6791142 (T/C) and rs35874192 (G/C) showed allelic effects on transcriptional activities in both hNPCs and SH-SY5Y cell lines, with SCZ-associated allele C corresponding to a lower gene expression (Fig. 1a, b). Notably, the directions of allelic effects of rs6791142 and rs35874192 were consistent with eQTL patterns detected in brain tissues from the GTEx[24] and BrainSeq consortium[25] (Fig. 1a, b). While the eSNPs rs900818, rs7616204, rs570621, and rs7349597 had no differential allelic effects on transcriptional activities or showed inconsistent directions of allelic effects with the eQTL patterns (Fig. 1c–f).

### *PCCB* knockdown in hFOs affects expression of genes enriched in GABAergic synapse

According to the TWAS and DLRA results, lower *PCCB* expression is associated with increased SCZ risk. We established *PCCB* knockdown and control human induced pluripotent stem cells (hiPSCs, U2F) using CRISPR interference (CRISPRi). In CRISPRi, one guide RNA (gRNA) sequence targeting *PCCB* (*PCCB*-G1) and one nontargeting control gRNA were designed. The established *PCCB* knockdown and control U2F hiPSCs were then used to generate hFOs (Fig. 2a–d) to investigate the functional impacts of *PCCB* knockdown. On day 60 of organoid culture, *PCCB* knockdown and control hFOs were used for RNA-sequencing (RNA-seq). Differential gene expression analysis identified 2326 differentially expressed genes (DEGs) [false discovery rate (FDR) < 0.05] between the *PCCB* knockdown and control hFOs (Supplementary Data 2). Among the 2326 DEGs, 1099 genes were upregulated and 1227 genes were downregulated in *PCCB* knockdown hFOs (Fig. 2e, f).

We next used WebGestalt 2019[26] to annotate biological functions of the *PCCB* knockdown-induced DEGs. For the upregulated DEGs, Gene Ontology (GO) analysis revealed their significant enrichment in biological functions including RNA catabolic process (FDR < 2.20E-16), forebrain development (FDR = 1.11E-10), and neural precursor cell proliferation (FDR = 8.96E-09). The Kyoto Encyclopedia of Genes and Genomes (KEGG) pathway analysis implicated the upregulated DEGs in pathways including the ribosome (FDR < 2.20E-16) and hippo signaling (FDR = 2.43E-04) pathway (Fig. 2g and Supplementary Data 2). For the downregulated DEGs, GO analysis showed that they were enriched with neuronal functions, including synaptic vesicle cycle (FDR = 6.28E-10), neurotransmitter transport (FDR = 1.23E-09), and synapse organization (FDR = 2.27E-07). The KEGG pathway analysis implicated the downregulated DEGs in the gamma-aminobutyric acid (GABA)-ergic synapse (FDR = 1.73E-04), morphine addiction (FDR = 1.73E-04), and nicotine addiction (FDR = 2.31E-03) pathways (Fig. 2h and Supplementary Data 2). These results highlighted that reduced expression of *PCCB* may downregulate genes related to neuronal functions, including the GABAergic synapse.

To confirm the RNA-seq results and to minimize potential off-target effects in CRISPRi, we generated *PCCB* knockdown U2F hFOs using a second gRNA (*PCCB*-G2) and performed RNA-seq (Supplementary Fig. S2). We found that 1079 of 2326 DEGs in *PCCB*-G1 hFOs overlapped with those identified in *PCCB*-G2 hFOs. The 1079 overlapped DEGs were also enriched with neuronal functions such as synapse organization (FDR = 5.66E-04), forebrain development (FDR = 1.13E-02), and axon development (FDR = 1.51E-02) (Supplementary Fig. S2 and Supplementary Data 2). Of the 1079 overlapped DEGs, 350 genes were downregulated in both *PCCB*-G1 and *PCCB*-G2 hFOs. Using the shared 350 downregulated DEGs, we constructed a protein-protein interaction (PPI) network and found that GABA receptor genes including *GABRA1*, *GABRA2*, *GABRB2*, and *GABRB3* were hub nodes in the PPI network (Fig. 2i). Real-time quantitative PCR (RT-qPCR) further confirmed the decreased expression of these GABA receptor genes in *PCCB* knockdown U2F hFOs (Fig. 2j). These results further highlighted that *PCCB* knockdown may affect the GABAergic pathways.

### *PCCB* knockdown-induced DEGs in hFOs are enriched with SCZ-related genes

To explore *PCCB*'s connection to SCZ, we evaluated the enrichment of 1079 *PCCB* knockdown-induced DEGs in hFOs with SCZ-related gene sets. The first SCZ gene set was 4096 differentially expressed protein-coding genes between postmortem brains of 559 SCZ patients and 936 controls from the PsychENCODE consortium[10]. The second SCZ gene set was 2809 DEGs between cerebral organoids (6 months) derived from eight SCZ patients and eight controls from the Kathuria et al. study[27] (Supplementary Data 2). We found that the 1079 *PCCB* knockdown-induced DEGs were significantly overlapped with genes dysregulated in PsychENCODE SCZ brains (Overlapped genes = 282, *P* = 5.84E-04) and SCZ patient-derived cerebral organoids (Overlapped genes = 255, *P* = 1.92E-14) (Supplementary Fig. S3a). We also found that the 1079 *PCCB* knockdown-induced DEGs were significantly ($P_{adjust}$ = 7.09E-03) overlapped with genes reported in SCZ GWAS from the FUMA analysis[28] (Supplementary Fig. S3b). Interestingly, the overlapped SCZ-related genes included sveral GABAergic synapse-related genes such as *GABRA1*, *GABRA2*, *GABRB2*, and *GABRB3* (Supplementary Data 2). These results suggested that *PCCB* may function through affecting the expression of SCZ-related genes, including the GABAergic synapse-related genes.

### Metabolomic analysis confirms the decreased GABA levels in *PCCB* knockdown hFOs

RNA-seq analysis revealed the effects of *PCCB* knockdown on GABAergic synapse, we further used metabolomic analysis to examine

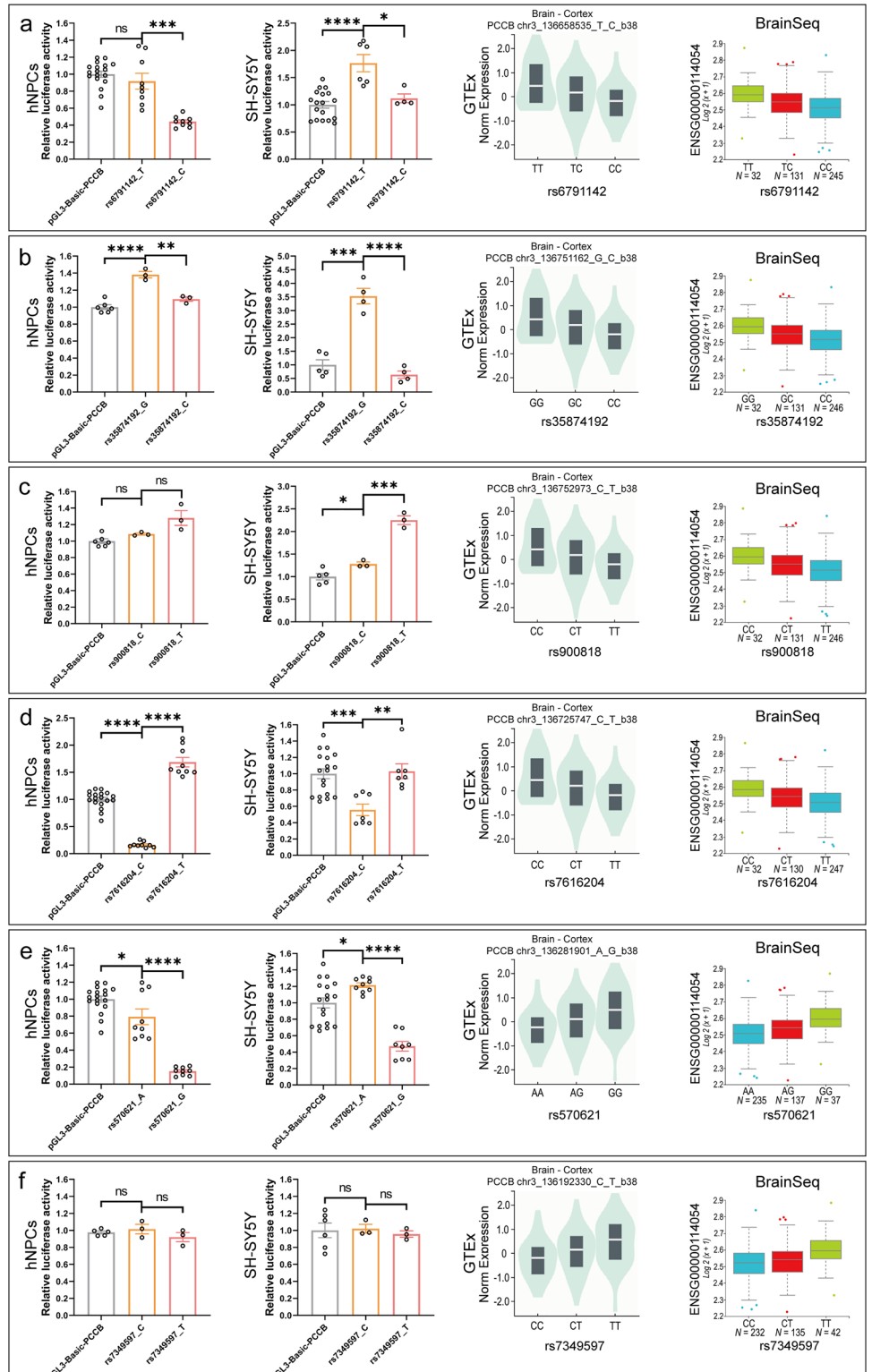

**Fig. 1 | DLRA results in hNPCs and SH-SY5Y cell lines. a–f** DLRA results for *PCCB* eSNPs rs6791142 (**a**), rs35874192 (**b**), rs900818 (**c**), rs7616204 (**d**), rs570621 (**e**), and rs7349597 (**f**) in hNPCs and SH-SY5Y cell lines. The PRL-TK Renilla vector was used as internal control. At least three biological replicates were used in per group. Data are shown as Mean ± SEM. Unpaired two tailed t-test was used for comparison between two groups. *$P < 0.05$, **$P < 0.01$, ***$P < 0.001$, ****$P < 0.0001$. eQTL plots in this figure were downloaded from the GTEx portal (https://www.gtexportal.org/home/) and BrainSeq phaseI eQTL data (http://eqtl.brainseq.org/phase1/eqtl/). Source data underlying a-f are provided as a Source Data file.

whether *PCCB* knockdown decreased GABA levels in hFOs. Of the 178 detected metabolites in U2F hFOs (day 60), 27 and 35 metabolites were differentially expressed in *PCCB*-G1 and *PCCB*-G2 hFOs (Fig. 3a–c and Supplementary Data 3), respectively. A total of 11 differential

metabolites were shared in both groups with 9 metabolites showing consistent alteration directions (Fig. 3c). The 9 shared differential metabolites were enriched in pathways including febrile seizures ($P = 1.96E-02$), GABA-transaminase deficiency ($P = 1.96E-02$), and

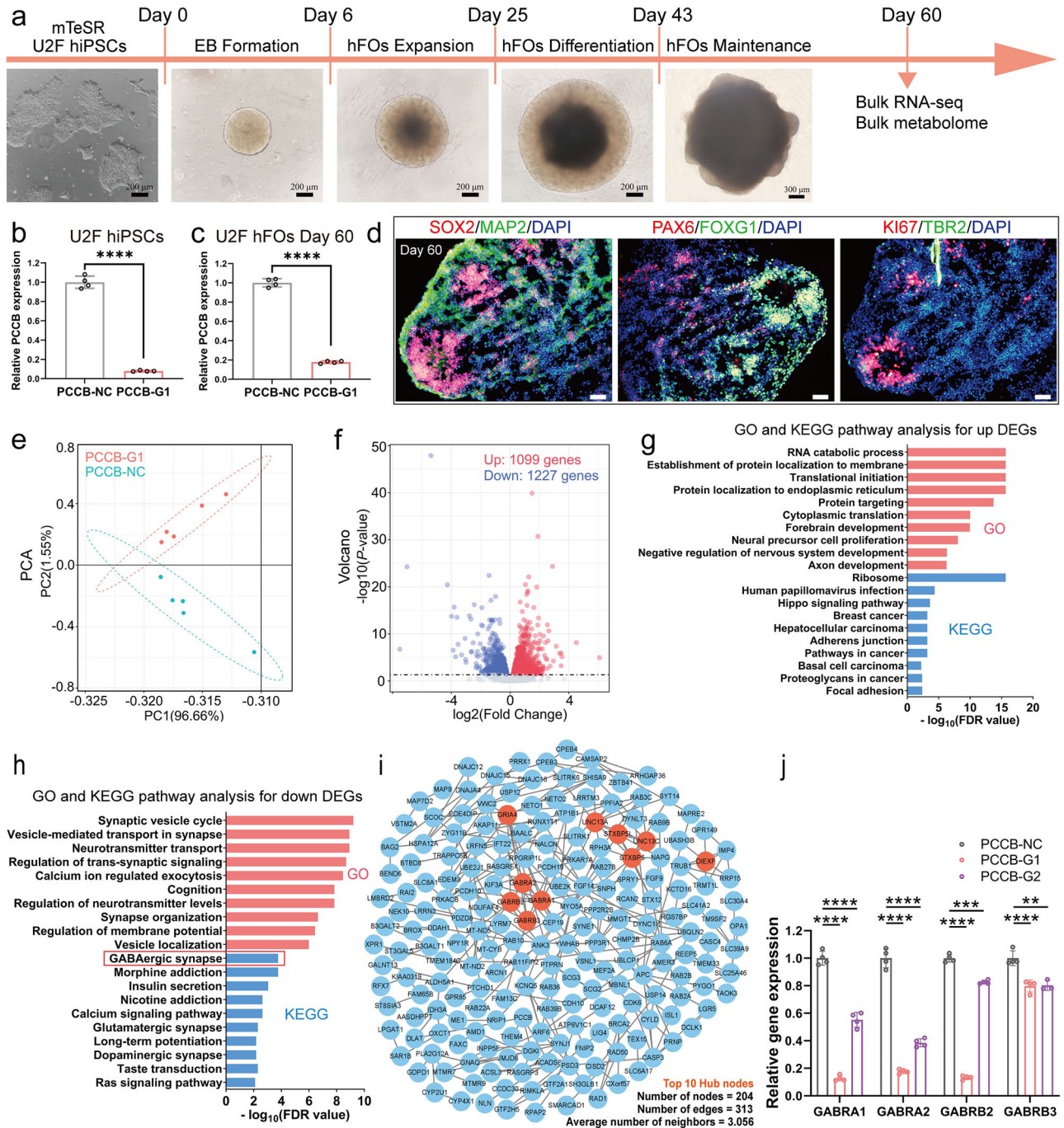

**Fig. 2 | Functional effects of *PCCB* knockdown in U2F hFOs. a** Workflow of hFOs culture in this study. Scale bar, 200 μm and 300 μm. **b, c** RT-qPCR analysis for *PCCB* expression in hiPSCs (**b**) and hFOs (**c**), *n* = four technical replicates per group. **d** Immunostaining characterization for representative control hFOs. *N* ≥ three biologically independent samples in each group. Cortical plate marker, MAP2; intermediate zone marker, TBR2; ventricular zone markers, SOX2 and ki67; forebrain-specific markers, FOXG1 and PAX6. Scale bar, 50 μm. **e** PCA plot for the *PCCB* knockdown and control hFOs. **f** Volcano plot of DEGs between the *PCCB* knockdown and control hFOs. Upregulated genes are shown with red dots and downregulated genes are shown with blue dots. **g, h** GO and KEGG analysis for the *PCCB* knockdown-induced upregulated (**g**) and downregulated DEGs (**h**), respectively. GO terms are shown with red bars, and KEGG pathways are shown with blue bars. **i** PPI network analysis for 350 downregulated DEGs shared in both *PCCB*-G1 and *PCCB*-G2 hFOs. The top 10 hub nodes are shown with orange nodes. **j** RT-qPCR analysis for *GABRA1*, *GABRA2*, *GABRB2*, and *GABRB3* (The hub nodes in the PPI network), *n* = four technical replicates per group. Data are shown as Mean ± SD. The unpaired two-tailed t-test was used to assess difference between the *PCCB*-NC and *PCCB*-G1 or *PCCB*-G2 group. **P < 0.01, ***P < 0.001, ****P < 0.0001. Source data underlying **b**, **c**, and **j** are provided as a Source Data file.

spinocerebellar degeneration (*P* = 2.93E-02) (Fig. 3d). Notably, GABA levels were decreased in both *PCCB*-G1 (decreased by 59%, *P* = 3.42E-03) and *PCCB*-G2 hFOs (decreased by 22%, *P* = 1.31E-02) when compared with control hFOs (Supplementary Data 3). The Pearson correlation analysis between the 9 shared differential metabolites and *PCCB*

knockdown-induced DEGs further confirmed a positive correlation (r = 0.51, *P* = 0.03) between *PCCB* expression and GABA. Notably, GABA showed strong positive correlations (r > 0.73, *P* < 5.50E-04) with GABA receptor genes *GABRA1*, *GABRA2*, *GABRB2* and *GABRB3*, respectively (Fig. 3e and Supplementary Data 3).

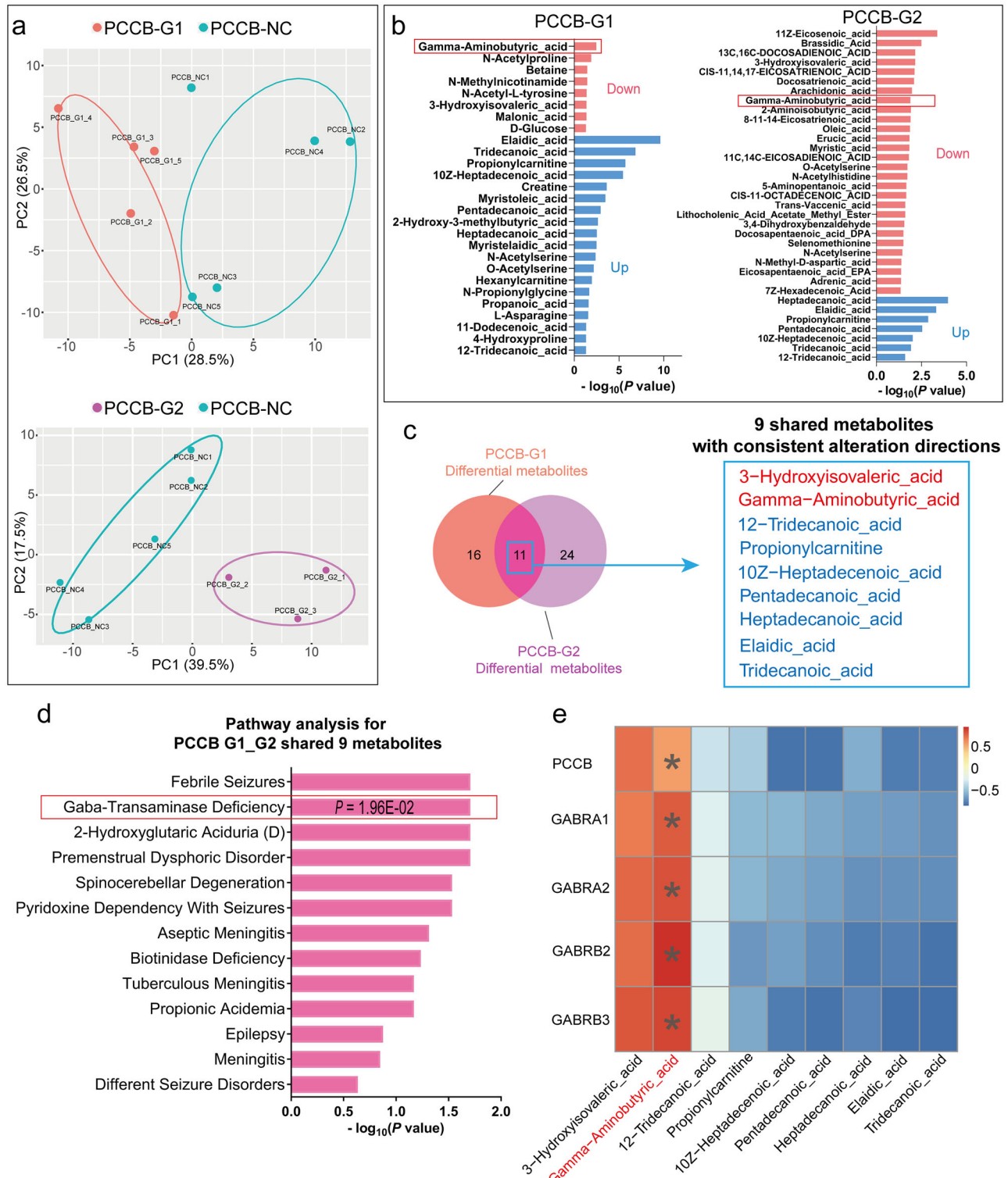

**Fig. 3 | Metabolomic analysis after *PCCB* knockdown in U2F hFOs. a** PCA plot for the metabolomic analysis in U2F hFOs. $N \geq$ three biologically independent samples in each group. **b** Metabolomic analysis identified 27 and 35 differentially expressed metabolites in *PCCB*-G1 and *PCCB*-G2 hFOs, respectively. GABA (indicated by red box) was downregulated in both groups. **c** Overlap for differential metabolites identified in *PCCB*-G1 and *PCCB*-G2 hFOs. **d** Functional annotation for 9 shared differential metabolites with consistent alteration directions in *PCCB*-G1 and *PCCB*-G2 hFOs. *P* values were calculated by the hypergeometric test. **e** Pearson correlation analysis between *PCCB* knockdown-induced DEGs and differential metabolites. The unpaired two-tailed t-test was used to calculated the *P* value. *$P < 0.05$. Their correlation coefficients are indicated by the color bar.

## *PCCB* knockdown in hFOs leads to mitochondrial dysfunction and reduction of GABA levels

*PCCB* encodes the β subunit of the propionyl-CoA carboxylase, a mitochondrial enzyme involved in the catabolism of propionyl-CoA[29].

*PCCB* mutation has been reported to impair mitochondrial energy metabolism by disrupting the tricarboxylic acid (TCA) cycle[16]. We would expect mitochondrial dysfunction caused by *PCCB* knockdown. Indeed, we found several mitochondrial genes that function in cellular

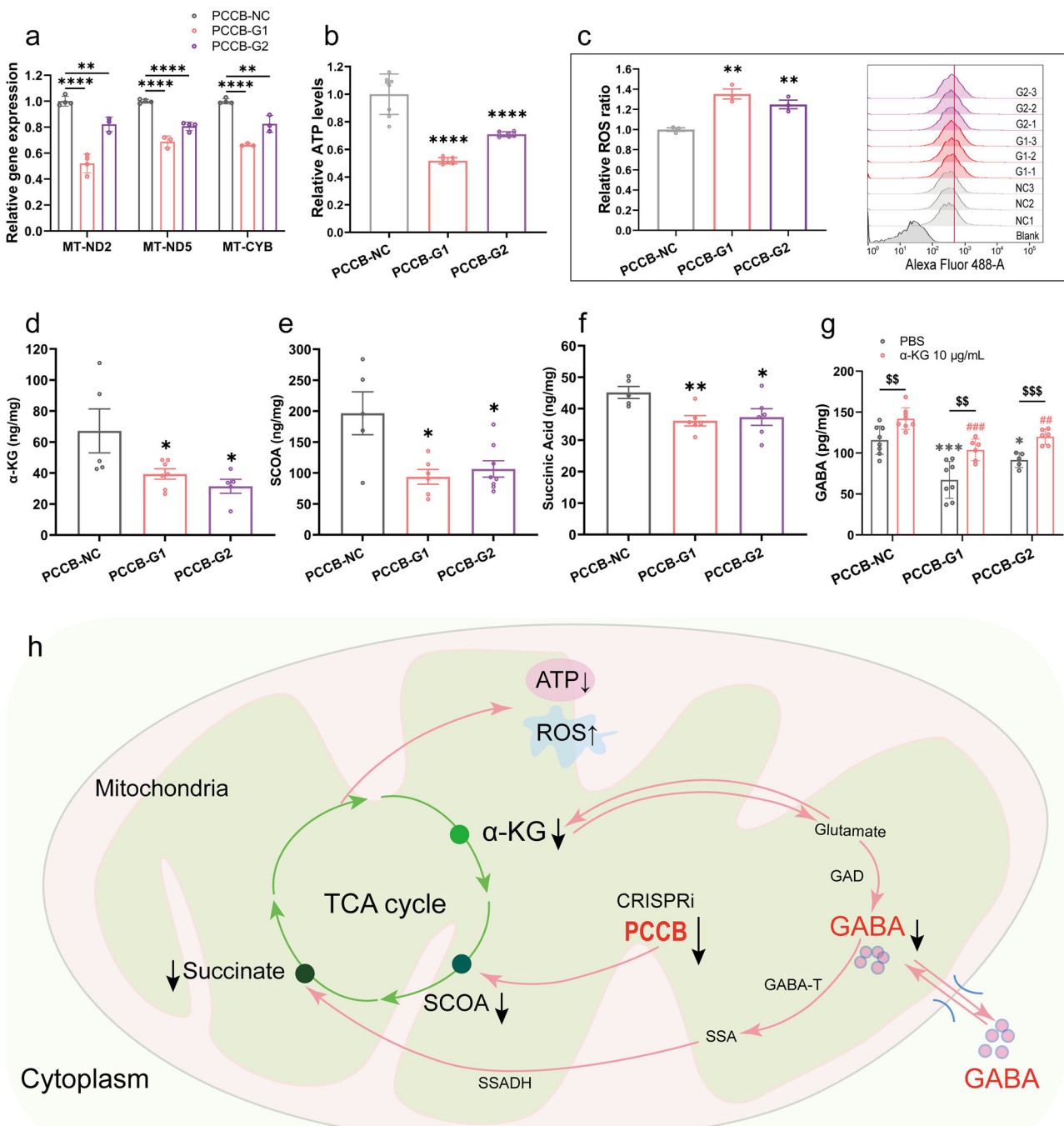

**Fig. 4 | *PCCB* knockdown leads to mitochondrial dysfunction and decreased GABA levels in U2F hFOs. a** RT-qPCR analysis revealed downregulated expression of mitochondrial genes in *PCCB* knockdown hFOs. **b, c** Decreased ATP (**b**) and increased ROS contents (**c**) were observed in *PCCB* knockdown hFOs. **d, e, f** ELISA analysis validated the reduction of α-KG (**d**), SOCA (**e**), and succinic acid (**f**) in *PCCB* knockdown hFOs. **g** Adding α-KG (10 μg/mL) into the culture medium of *PCCB* knockdown hFOs restored the GABA levels. **h** Potential pathways involved in the regulation of *PCCB* knockdown on GABA synthesis. GAD, glutamate decarboxylase;

GABA-T, GABA transaminase; SSA, Succinic semialdehyde; SSADH, succinic semi-aldehyde dehydrogenase; ↓, downregulation; ↑, upregulation. $N \geq$ three biologically independent samples in each group. Data are shown as Mean ± SEM. The unpaired two-tailed t-test was used to assess difference between the *PCCB*-NC and *PCCB*-G1 or *PCCB*-G2 group. *$P < 0.05$, **$P < 0.01$, ***$P < 0.001$, ****$P < 0.0001$, ##$P < 0.01$, ###$P < 0.001$, $$$P < 0.01$, $$$$P < 0.001$. Source data underlying a-g are provided as a Source Data file.

oxidative phosphorylation, including *MT-ND2*, *MT-ND5*, and *MT-CYB*, were downregulated in both *PCCB*-G1 and *PCCB*-G2 U2F hFOs (Supplementary Data 2), which was further validated by the RT-qPCR analysis (Fig. 4a). Adenosine triphosphate (ATP) and reactive oxygen species (ROS) detection assays showed that *PCCB* knockdown reduced ATP generation and increased ROS levels in U2F hFOs (Fig. 4b, c), indicating the mitochondrial dysfunction caused by *PCCB* knockdown.

Since GABA metabolism involves a route from α-ketoglutarate (α-KG) generated by the TCA cycle to succinate via glutamate, GABA, and succinic semialdehyde[30,31], we examined whether *PCCB* knockdown decreased GABA levels by inhibiting the TCA cycle. We performed enzyme linked immunosorbent assay (ELISA) and confirmed that *PCCB* knockdown decreased α-KG, succinyl-CoA (SCOA), and succinic acid (Fig. 4d–f), three key metabolites that connect the GABA shunt and

TCA cycle[30,31]. Considering that α-KG is an upstream metabolite that could be converted to GABA, we added α-KG (10 μg/mL) into culture media of U2F hFOs and found a restored GABA levels in *PCCB* knockdown hFOs (Fig. 4g).

To confirm the functional impacts of *PCCB* knockdown observed in U2F hFOs, we generated hFOs using another hiPSC line (U1M) (Supplementary Fig. S4a, b). The RT-qPCR and mitochondrial functional analyses showed that *PCCB* knockdown in U1M hFOs also resulted into decreased expression of mitochondrial genes (*MT-ND2*, *MT-ND5* and *MT-CYB*), reduced ATP generation and increased ROS levels (Supplementary Fig. S4c-e). The ELISA showed that *PCCB* knockdown decreased α-KG, SCOA, succinic acid, and GABA levels in U1M hFOs (Supplementary Fig. S4f-i). Adding α-KG (10 μg/mL) into culture media of U1M hFOs restored the GABA levels (Supplementary Fig. S4i) as detected in U2F hFOs. These results indicated that *PCCB* knockdown decreased GABA levels by inhibiting the TCA cycle and led to mitochondrial dysfunction (Fig. 4h).

### *PCCB* knockdown in hFOs leads to abnormal electro-physiological activities

Since GABA, the major inhibitory neurotransmitter in the brain[32], was decreased in *PCCB* knockdown hFOs, we used multielectrode array (MEA) recording assay to test whether *PCCB* knockdown affected neuroactivities. The *PCCB* knockdown and control U2F hFOs at day 160 were seeded on Matrigel-coated 24-well MEA plate (Fig. 5a). After 7 days of culture, electroactivities of hFOs were recorded (Fig. 5b−d).

We found that *PCCB* knockdown in hFOs led to increased number of spikes (Fig. 5e) and mean neuron firing rate (Fig. 5f), suggesting a hyper neuroactivity after *PCCB* knockdown in U2F hFOs. However, *PCCB* knockdown decreased the synchronization of the neural network (Fig. 5g). These MEA results were further replicated in U1M hFOs (Supplementary Fig. S5). Since hyper neuroactivity and decreased synchronization of neural network in SCZ brains have been reported by the electroencephalography and magnetoencephalography[33], these results supported that *PCCB* knockdown led to abnormal electro-physiological activities that link to SCZ phenotypes.

## Discussion

SCZ is a polygenic psychiatric disorder with risk contributed by multiple genes. Identifying genes whose expression is associated with SCZ risk by TWAS is a powerful approach to prioritize SCZ risk genes. By integrating multiple published datasets from TWAS, gene coexpression, and differential gene expression analysis, we prioritized *PCCB* as a reliable SCZ risk gene. *PCCB* is a gene encoding β subunit of propionyl-coA carboxylase enzyme[29], defect of which has been reported as a cause of propionic acidemia[34]. Though several case reports have shown neuropathological symptoms including autistic features[35] in propionic acidemia[36,37], how *PCCB* deficiency led to neuropathology is largely unknown. Moreover, what roles *PCCB* plays in the etiology of SCZ has not been investigated.

To investigate *PCCB*'s contribution to SCZ risk, we performed RNA-seq analysis and identified that *PCCB* knockdown in U2F hFOs

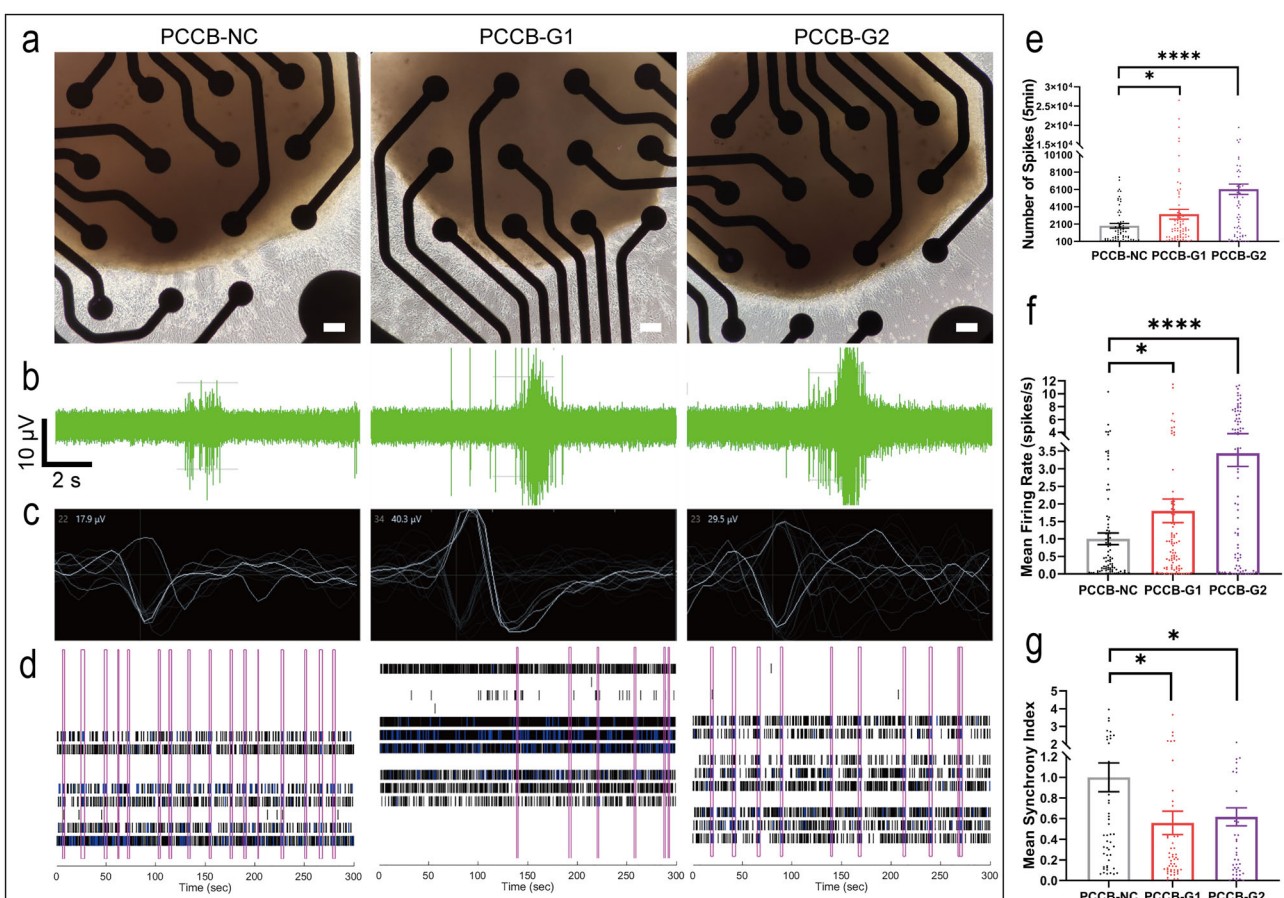

**Fig. 5 | MEA assay after *PCCB* knockdown in U2F hFOs. a** Bright field of hFOs cultured in 24-well MEA plate. Scale bar, 100 μm. **b** Representative burst traces for individual electrode recorded from hFOs. **c** Schematic diagram of a single unit event. **d** Raster plot of neural network activity. Each pink box represents a synchronized burst. **e**−**g** *PCCB* knockdown increased the number of spikes (**e**) and mean neuron firing rate (**f**) but reduced the synchronized burst index in hFOs (**g**). N ≥ 13 biologically independent samples in each group. Data are shown as Mean ± SEM. The unpaired two-tailed t-test was used to assess difference between the *PCCB*-NC and *PCCB*-G1 or *PCCB*-G2 group. *P < 0.05, ****P < 0.0001. Source data underlying e-g are provided as a Source Data file.

downregulated expression of genes related to multiple neuronal functions including the GABAergic synapse pathway. Several GABA receptor genes including *GABRA1*, *GABRA2*, *GABRB2*, and *GABRB3* were highlighted as hub genes in the *PCCB*'s regulatory network (Fig. 2i). To confirm these results, we generated *PCCB* knockdown hFOs using two other hiPSC lines (U1M and ACS-1011) (Supplementary Fig. S6a, b). The RT-qPCR analysis confirmed that *PCCB* knockdown also led to decreased expression of *GABRA1*, *GABRA2*, *GABRB2*, and *GABRB3* in both U1M and ACS-1011 hFOs (Supplementary Fig. S6c, d).

The downregulated GABAergic synapse pathway caused by *PCCB* knockdown attracted our attention, since GABAergic dysfunction plays important roles in SCZ etiology[38,39]. To validate the GABA findings in this study, we used immunostaining assay and confirmed the existence of cells expressing the GABAergic markers GAD1 (GAD67) and DLX2 in hFOs (Supplementary Fig. S7). We further performed metabolomic analysis and confirmed the decreased GABA levels in *PCCB* knockdown hFOs. The following electrophysiological analysis showed that *PCCB* knockdown in both U1M and U2F led to hyper neuroactivity and decreased synchronization of neural network activities, cellular phenotypes reported to be associated with SCZ risk[33,40]. Through the hFOs-based multiomics analyses, we revealed the functional impacts of *PCCB* knockdown and highlighted that *PCCB* dysregulation may contribute to SCZ etiology through regulating the GABAergic pathways.

Since the GABA shunt connected the GABA metabolism pathway and TCA cycle[30,31], we expected that *PCCB* regulates GABAergic pathways by affecting the TCA cycle. As expected, *PCCB* knockdown led to reduced production of α-KG, SCOA, and succinic acid in the TCA cycle. Since α-KG produced from the TCA cycle could be converted into SCOA or serve as a source for GABA synthesis[41], the decrease of α-KG may be responsible for the reduced production of GABA. On the other hand, *PCCB* knockdown leads to the reduction of SCOA and succinic acid, which may exacerbate the entry of GABA into the TCA cycle through GABA shunt pathway and further reduce GABA content in the cytoplasm[41]. Overall, *PCCB* knockdown decreased GABA levels by reducing the content of α-KG, SCOA and succinic acid in the TCA cycle (Fig. 4h). Our study highlighted that mitochondrial dysfunction caused by *PCCB* knockdown not only impaired cellular energy production, but also resulted into GABAergic dysfunction. Since mitochondrial dysfunction is associated with GABAergic dysfunction[42] and the etiology of SCZ[43,44], our study connected the "mitochondrial hypothesis" and "GABA hypothesis" of SCZ.

In addition to mitochondrial dysfunction, one of the major effects of *PCCB* defect is the cellular accumulation of propanoic acid, propionyl carnitine, and other metabolites[16]. Indeed, we did observe a dramatic increase of propanoic acid or propionyl carnitine in *PCCB* knockdown hFOs (Supplementary Data 3). Interestingly, hFOs exposed to propanoic acid at 3.2 μM, an average serum concentration in SCZ patients[45], led to decreased expression of GABA receptor genes (*GABRA1*, *GABRA2*, *GABRB2*, and *GABRB3*) (Supplementary Fig. S8), which is consistent with those observed in *PCCB* knockdown hFOs. These results suggested that the accumulation of propanoic acid may mediate the effects of *PCCB* knockdown, and partially explain how propionic acidemia could lead to neuropathological symptoms. These results also suggested the potential effects of short-chain fatty acids on SCZ risk, since short-chain fatty acids including propanoic acid, acetic acid, and butyric acid were found to be upregulated in the serum of SCZ patients[46].

This study reveals the connection between *PCCB* and SCZ risk, some limitations also exist. First, *PCCB* knockdown affects multiple types of synapses, including GABAergic, glutamatergic, dopaminergic, and cholinergic synape, as revealed by bulk RNA-seq analysis. The cell-type specific effects of *PCCB* knockdown is unclear. Further investigations such as RNA-seq and other omics analysis at single-cell level are

needed. Second, hFOs (day 60) are immature in this study since the average proportion of GAD67+ or DLX2+ GABAergic neurons only account for about 5% of the total cells in hFOs (Supplementary Fig. S7). A long-term organoid culture could allow hFOs develop further to generate abundant GABAergic neurons to validate the GABA findings. Last, we showed that SCZ-associated SNPs rs6791142 and rs35874192, two eQTL SNPs for *PCCB*, affected transcriptional activities. But whether these two eSNPs could affect *PCCB* expression in vivo remains unclear. In the future, using CRISPR-Cas9 gene editing to confirm the regulatory effects of eSNPs rs6791142 and rs35874192 on *PCCB* expression is needed.

In summary, this study used hFOs-based multiomics analyses and revealed connection between *PCCB* and SCZ, highlighting that *PCCB* may contribute to SCZ etiology through regulating the GABAergic pathways and mitochondrial function.

## Methods

### Prioritization of SCZ risk genes and SNPs

We collected the published data from TWAS[8–11], MR-JTI[15], and SMR[2,10,12,13] analyses to prioritize SCZ risk genes with sufficient supporting evidence. We also checked whether the prioritized genes are located in SCZ risk-associated gene coexpression modules or differentially expressed in postmortem brains of SCZ patients.

To prioritize SCZ risk SNPs, we first collected the top SNPs from the TWAS data. We then retrieved SNPs in LD ($r^2 \geq 0.6$, European population genome) with the top SNPs. We prioritized candidate causal SNPs that likely affected gene expression in the brain using the following criteria: 1) candidate SNPs are eSNPs for the SCZ risk genes in the brain based on the BrainSeq[25], GTEx[24], or PsychENCODE eQTL data[47]; 2) SNPs are located within chromatin open regions based on ATAC-seq data from the PsychENCODE consortium; 3) SNPs are located in genomic regions predicted as enhancers or promoters in human brain tissues or neural cell cultures based on the Roadmap Epigenomics data[23].

### Cell culture

Three hiPSC lines (U1M, U2F, and ACS-1011) used in this study were derived from healthy individuals. The U1M and U2F hiPSCs were obtained from the Cellapy Technology (Beijing, China), and the ACS-1011 cell line was obtained from the American type culture collection. Pluripotency and karyotype of hiPSCs were confirmed by immuno-fluorescence and karyotype analysis as shown in our previous studies[48,49]. The hiPSCs were cultured in Matrigel (Corning, 354277)-coated plate and supplemented with the mTeSR Plus medium (STEMCELL Technologies, 05825) and 1% penicillin/streptomycin (Gibco, 10378016).

hNPCs were induced from U2F hiPSCs using the STEMdiff™ Neural Induction Medium (STEMCELL Technologies, 05835) as described in our previous study[48]. In brief, hiPSCs were dissociated into single cells using Accutase solution (Sigma-Aldrich, A6964) on day 0. Cells were then cultured in Matrigel-coated12-well plate and fed with 600 μL Neural Induction Medium supplemented with 1x penicillin/streptomycin and 10 μM Y27632 (Selleck, SCM075). On day 3, each well was fed with 600 μL fresh Neural Induction Medium without Y27632. From day 18, the hNPCs were cultured and maintained in the STEMdiff™ Neural Progenitor Medium (STEMCELL Technologies, 05833). The culture medium was completely replaced with fresh medium every two days.

The SH-SY5Y neuroblastoma cells were cultured in high-glucose DMEM (Gibco, C11995500BT) supplemented with 10% fetal bovine serum (Gibco, A3161001C) and 1% penicillin/streptomycin.

### DLRA

About 50 bp DNA sequence (Supplementary Data 4) flanking the *PCCB* eSNP was synthesized and cloned into the pGL3-basic vector using the

restriction enzymes KpnI and XhoI (New England BioLabs). The *PCCB* promoter sequence (~600 bp) was amplified from the genomic DNA using the PCR primers (F, 5′-CCGCTCGAGTTTGAATCCTGGCCAAC CAC-3′; R, 5′-CCC AAGCTTTGCTAAAGCGTGGGTACGG-3′) and Phanta®Max Super-fidelity DNA Polymerase (Vazyme, P505-d1). The amplified *PCCB* promoter was then cloned into downstream of the eSNP-containing DNA fragment using the restriction enzymes XhoI and Hind III (New England BioLabs). DLRA was performed in both hNPCs and SH-SY5Y cells. For DLRA, $1 \times 10^5$ cells per well were plated into 24-well plate. After 24 hours of culture, 500 ng recombinant pGL3-basic luciferase reporter vector and 20 ng PRL-TK Renilla internal control vector for each well were co-transfected into cells using Lipofectamine™ 3000 (Invitrogen, L3000015). 36 h post transfection, the Firefly and Renilla luciferase activities were measured on the LumiPro luminescence detector (Lu-2021-C001) using the DLRA kit (Vazyme, DL101-01). Experiments were conducted in at least three biological replicates.

### Establishment of *PCCB* knockdown and control hiPSCs

We used CRISPRi to establish *PCCB* knockdown and control hiPSCs. Two gRNA sequences targeting *PCCB* (*PCCB*-G1, 5′-GCATTACGGG TGGCGGCGGT-3′; *PCCB*-G2, 5′-GCGTACTCAGGTGCGCCGGT-3′) were designed using the online tool CRISPR-ERA (http://crispr-era.stanford.edu/). The *PCCB*-targeting gRNA or control gRNA sequence (5′-GCGCCAAACGTGCCCTGACGG-3′)[50] were synthesized and cloned into the lentiviral vector pLV-hU6-sgRNA-hUbC-dCas9-KRAB-T2a-Puro. The constructed vectors were then used for lentiviral package (ObiO Technoliges, China). For viral infection in hiPSCs, when cell confluence reached 40–50% in 12-well plate, cells were cultured in 0.6 mL medium and incubated with the lentiviruses for 24 h. The culture medium was fully replaced in the next day. 48 h post viral infection, cells were treated with 0.5–1 μg/mL puromycin for 3–7 days to kill cells that were not infected by the lentiviruses. *PCCB* knockdown efficiency was confirmed by RT-qPCR analysis.

### RT-qPCR analysis

Total RNA was used to generate complementary DNA using HiScript III RT SuperMix for qPCR (+gDNA wiper) (Vazyme, R323-01). RT-qPCR assay was performed using ChamQ SYBR qPCR Master Mix (Vazyme, Q711-02) on the Real-Time PCR System (Roche, LightCycler 480 II). GAPDH was used as the internal reference gene. At least three technical replicates were used in the RT-qPCR analysis. RT-qPCR primers are provided in Supplementary Data 4.

### Generation of hFOs

The established *PCCB* knockdown and control hiPSCs were used to generate hFOs using the STEMdiff Dorsal Forebrain Organoid Differentiation Kit (STEMCELL Technologies, 08620) based on the manufacturer's instructions with some modifications. Briefly, hiPSCs were dissociated into single cells using Accutase solution (Sigma-Aldrich, A6964) on day 0. $1 \times 10^4$ cells per well were then plated into 96-well round-bottom ultra-low attachment plate (Corning, 7007) and fed with 50 μL Forebrain Organoid Formation Medium supplemented with 1x penicillin/streptomycin and 10 μM Y27632 (Selleck, SCM075). On day 3, each well was gently added with 50 μL fresh Forebrain Organoid Formation Medium without Y27632. On day 6, the medium was replaced with the Forebrain Organoid Expansion Medium. On day 25, the Forebrain Organoid Expansion Medium was replaced with the Forebrain Organoid Differentiation Medium. From day 43, organoids were cultured in the Forebrain Organoid Maintenance Medium. hFOs were characterized using immunostaining analysis.

### Immunostaining analysis

hFOs were sectioned at a thickness of 16 μm. hFOs sections were treated with 1% Triton X-100 for 30 mins. After washing with PBS three

times, hFOs sections were blocked with 5% bovine serum albumin (Solarbio, A8020) for 1 h at 37 °C. The hFOs sections were then incubated with primary antibodies overnight at 4 °C. After washing with PBS three times, hFOs sections were incubated with secondary antibodies for 1 h and DAPI (10 μg/ml Invitrogen, D1306) for 5 mins at room temperature. Primary antibodies used in this study were mouse anti-Ki67 (1:1,000, Cell Signaling Technologies, 9449), rabbit anti-FOXG1 (1:100, Abcam, ab18259), rabbit anti-TBR2 (1:800, Cell Signaling Technologies, 81493), mouse anti-SOX2 (1:200, Invitrogen, MA1-014), rabbit anti-PAX6 (1:100, Proteintech, 12323-1-AP), rabbit anti-MAP2 (1:200, Proteintech, 17490-1-AP), mouse anti-GAD67 (1:200, Abcam, ab26116) and mouse anti-DlX2 (1:200, Santa cruz, sc-393879). The secondary antibodies were Alexa Fluor 594-conjugated goat anti-mouse IgG (1:300, Abcam, ab150116), Alexa Fluor 594-conjugated goat anti-rabbit IgG (1:200, Invitrogen, A-11012), Alexa Fluor 488-conjugated goat anti-rabbit (1:200, Invitrogen, A-11008), goat anti-mouse Cy3 (1:200, BOSTER, BA1031) and goat anti-mouse CoraLite488 (1:200, Proteintech, SA00013-1). All antibodies used in this study were also shown in Supplementary Data 4.

### Bulk organoid RNA-seq and data analysis

Two hFOs from *PCCB* knockdown or control group were randomly selected and pooled together as one mixed sample. Five mixed samples in each group were used for total RNA extraction using the miRNeasy Mini Kit (Qiagen, 217004). RNA quality was evaluated on the Agilent 2100 Bioanalyzer system. RNA samples with RNA integrity numbers over 7 were used for RNA-seq (150 bp, paired-end) on the Illumina NovaSeq 6000 system.

Raw RNA-seq data were filtered to get clean reads using FastQC (v0.20.0). The clean reads were aligned to the human genome hg38 using STAR (v2.7.9a). Gene expression quantification was conducted using RSEM (v1.3.0) based on Gencode v40 comprehensive gene annotation. The filterByExpr function in the edgeR package (v3.36.0) was used to filter out low-expression genes. The sva function in the SVA package (v3.42.0) was used to estimate batch effect and other artifacts. Differential gene expression analysis between the *PCCB* knockdown and control group was performed using the DEseq2 package (v1.34.0). *P* values were adjusted using the Benjamini–Hochberg method. To annotate the functions of DEGs, we used the online tool WebGestalt2019 (http://www.webgestalt.org/) to perform GO and KEGG pathway enrichment analysis.

### PPI analysis

The STRING database (v11.5) (http://www.string-db.org/) was used to construct a high-confidence (interaction score > 0.7) PPI network for the *PCCB* knockdown-induced DEGs. Active interaction sources included text-mining, experiments, databases, coexpression, neighborhood, gene fusion, and co-occurrence. The PPI network was visualized using Cytoscape (v 3.9.1). CytoHubba[51], a Cytoscape plugin, was used to explore hub nodes in the PPI network.

### Metabolomic analysis of bulk hFOs

For metabolomic analysis, three hFOs (day 60) in *PCCB* knockdown or control group were randomly selected and pooled together as one mixed sample. At least three mixed samples in each group were then used for HM400 metabolomic analysis (Beijing Genomics Institute, China). Briefly, hFOs or quality control samples were lysed in 140 μL 50% water/methanol solution. The lysate was centrifuged (18,000 g, 4 °C, 20 min) to get the supernatant. The supernatant was used for derivatization reaction and then centrifuged at 4000 g, 4 °C, 10 min. The supernatant was further used for high performance liquid chromatography tandem mass spectrometer (LC-MS/MS) analysis on the SCIEX QTRAP 6500 + LC-MS/MS system. Parameters of liquid chromatographic column were BEH C18 (2.1 mm×10 cm, 1.7 μm, Waters).

The parameter of mass spectrometry was ESI + /ESI·. Content of metabolites (μmol/g) was quantified using the HMQuant software based on the formula (C*0.14/m), where C represents the calculated concentration (μmol/L) and m represents the sample weight (mg). Two-tailed t-test was used to identify differential metabolites between *PCCB* knockdown and control hFOs. Functional annotation for the differentially expressed metabolites was performed using the online tool MetaboAnalyst 5.0 (https://www.metaboanalyst.ca/).

## ATP assay

The ATP Assay Kit (Beyotime, S0026) was used to examine the effects of *PCCB* knockdown on ATP production in hFOs. At day 60, *PCCB* knockdown or control hFOs were washed twice with PBS and lysed with 200 μL lysis buffer per well. The lysate was centrifuged at 12,000 g for 5 min at 4 °C. The supernatant was used to measure ATP content on the LumiPro luminescence detector (Lu-2021-C001). Experiments were conducted in at least three biological replicates.

## ROS assay

The ROS Assay Kit (Beyotime, S0033S) was used to measure ROS levels in *PCCB* knockdown and control hFOs. Briefly, hFOs (day 60, three hFOs were randomly pooled together as one mixed sample) were dissociated into single cells using Accutase. After centrifuging at 1500 rpm for 5 min, cells were resuspended with prewarmed serum-free DMEM-F12 medium. The collected cells were incubated with 10 μM DCFH-DA fluorescent probes in serum-free DMEM-F12 for 20-30 min at 37 °C, 5% $CO_2$. $1 \times 10^4$ cells were analyzed on the flow cytometry (BD FACSAria II). Experiments were conducted in at least four biological replicates.

## ELISA assay

The ELISA kits (CAMILO, Nanjing, China) were used to measure SOCA (Cat No. 2H-KMLJh315292), succinic Acid (Cat No. 2H-KMLJh313077), α-KG (Cat No. 2H-KMLJh313735) and GABA (Cat No. 2H-KMLJh310295) levels in *PCCB* knockdown and control hFOs according to the manufacturer's instructions. Briefly, protein was extracted from hFOs (day 60, two hFOs were randomly pooled together as one mixed sample) using RIPA buffer (Beyotime, P0013B) containing 1% protease inhibitor. The lysate was centrifuged at 12,000 g for 10 min at 4 °C. Protein concentration was determined using the BCA protein assay kit (Vazyme, E112-01). The supernatant was used to detect SCOA, succinic Acid, α-KG and GABA. We also used ELISA to determine the GABA levels in *PCCB* knockdown hFOs after incubating the hFOs with 10 μg/mL exogenous α-KG[52] for 3 h at 37 °C, 5% $CO_2$. Experiments were conducted in at least four biological replicates.

## MEA assay

We used an MEA assay to evaluate the effects of *PCCB* knockdown on electrophysiological properties in hFOs. Briefly, hFOs were cultured in the Matrigel-coated 24-well MEA plate with one organoid seeded in each well. After 7 days of culture, electro activities were recorded using the Axion Biosystems for 7 days. Each recording duration was 5 min. The MEA data including the number of spikes, burst frequency, and network synchronization were analyzed using the software Axis Navigator 3.6.2.2.

## Statistical analysis

Data were analyzed with GraphPad Prism 9.0.0 and shown as Mean ± SEM. A two-tailed t-test was used to assess the difference between two groups. The hypergeometric test was used to assess the enrichment of *PCCB* knockdown-induced DEGs with SCZ-related gene sets. *P* values in the differential gene expression analysis were adjusted for multiple testing using the Benjamini–Hochberg method. For all statistical analyses, a *P* value less than 0.05 is considered statistically significant.

## Reporting summary

Further information on research design is available in the Nature Portfolio Reporting Summary linked to this article.

## Data availability

RNA-seq data generated in this study have been deposited in the Gene Expression Omnibus under accession code GSE226233. Because the raw data of the MEA and metabolomic data are huge and presented in highly diverse nature and formats, these raw data are available on request from the corresponding authors. All the other data associated with this study are shown in the manuscript, Supplementary Information, and Source Data file. Source data are provided with this paper. Public database: STRING (https://cn.string-db.org/), WebGestalt (https://www.webgestalt.org/option.php), MetaboAnalyst 5.0 (https://www.metaboanalyst.ca/). Source data are provided with this paper.

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

## Acknowledgements

This work was supported by grants from the National Natural Science Foundation of China (Nos. 82171504, 82022024, 31970572, and 81901359), the Natural Science Foundation of Hunan Province (No. 2022JJ20035), the Science and Technology Innovation Program of Hunan Province (2022RC1214), the Innovation-Driven Project of Central South University (No. 2020CX003), and the National Institute of Health (No. 1U01MH116489). ATAC-seq data were generated as part of the PsychENCODE Consortium (U01MH103392, U01MH103365, U01MH103346, U01MH103340, U01MH103339, R21MH109956, R21MH105881, R21MH 105853, R21MH103877, R21MH102791, R01MH111721, R01MH110928, R01MH110927, R01MH110926, R01MH110921, R01MH110920, R01MH 110905, R01MH109715, R01MH109677, R01MH105898, R01MH105898, R01MH094714, and P50MH106934).

## Author contributions

Q.M., C.C., and C.L. supervised the overall study, as well as guided the procedures and analyses. Q.M. and W.Z. wrote the manuscript. W.Z. did the CRISPRi experiment and organoid culture, as well as the mitochondrial function assay, ELISA, and MEA assay. M.Z. and H.Y. performed the differential gene expression analysis. Z.X., and H.W. performed the metabolomic and RT-qPCR analysis. B.T., C.C., C.L., J.J., and J.W. revised the manuscript.

## Competing interests

The authors declare no competing interests.

## Additional information

[1]Center for Medical Genetics & Hunan Key Laboratory of Medical Genetics, School of Life Sciences, and Department of Psychiatry, The Second Xiangya Hospital, Central South University, 410008 Changsha, Hunan, China. [2]The First Affiliated Hospital, Multi-Omics Research Center for Brain Disorders, Hengyang Medical School, University of South China, 421000 Hengyang, Hunan, China. [3]The First Affiliated Hospital, Clinical Research Center for Immune-Related Encephalopathy of Hunan Province, Hengyang Medical School, University of South China, 421000 Hengyang, Hunan, China. [4]The First Affiliated Hospital, Department of Neurology, Hengyang Medical School, University of South China, 421000 Hengyang, Hunan, China. [5]Department of Neurology, Xiangya Hospital, Central South University, 410008 Changsha, Hunan, China. [6]National Clinical Research Center for Geriatric Disorders, Xiangya Hospital, Central South University, 410008 Changsha, Hunan, China. [7]Department of Psychiatry, SUNY Upstate Medical University, Syracuse, NY 13210, USA. [8]Hunan Key Laboratory of Animal Models for Human Diseases, Central South University, Changsha, Hunan 410008, China. [9]Hunan Key Laboratory of Molecular Precision Medicine, Central South University, Changsha, Hunan 410008, China. [10]MOE Key Lab of Rare Pediatric Diseases & School of Life Sciences, University of South China, 421001 Hengyang, Hunan, China. ✉e-mail: liuch@upstate.edu; chenchao@sklmg.edu.cn; mengqingtuan@glmc.edu.cn

