## [Peer Review File · Nature Communications]

Human forebrain organoid-based multi-omics analyses of PCCB as a schizophrenia associated gene linked to GABAergic pathwaysREVIEWER COMMENTS

Reviewer #1 (Remarks to the Author):

Major comments:

1. In this study, the authors prioritized six SCZ-associated SNPs that may affect PCCB expression. Only three of these six SNPs were chosen for dual luciferase reporter assay. The remaining three SNPs could also be used for the dual luciferase reporter assay.
2. In lines 184-186, the authors should indicate which figure or table was used to show the metabolomic results “Notably, GABA levels were decreased in both PCCB- G1 (decreased by 59%, $P = 3.42E-03$) and PCCB-G2 hFOs (decreased by 25%, $P = 8.62E-185$) when compared with control hFOs”. On the other hand, ELISA should be used to validate the decreased GABA levels in PCCB knockdown hFOs. hFOs derived from another iPS cell line are also needed to validate these results.
3. In this study, the authors performed transcriptomic and metabolomic analysis after PCCB knockdown in hFOs. The correlations between the differentially expressed genes and metabolites should be analyzed.
4. As shown in the line 181, 14 differentially expressed metabolites were shared between PCCB-G1 and PCCB-G2 hFOs. The authors should indicate clearly how many metabolites were consistently altered in both PCCB-G1 and PCCB-G2 hFOs.

Minor comments :

1. There is a typo error in line 46, “deceased” should be corrected to “decreased”.
2. In lines 42-45, the sentence “PCCB knockdown in human forebrain organoids (hFOs) followed by RNA-seq revealed dysregulation of genes enriched with multiple neuronal functions including gamma-aminobutyric acid (GABA)-ergic synapse, as well as genes dysregulated in postmortem brains of SCZ patients or in cerebral organoids derived from SCZ patients” should be re-written for more clear understanding.
3. In line 110, there should be an abbreviation for hNPCs.
4. In Figure 2A, there are no scale bars in two figures of hFOs.
5. In the manuscript, “GABA level” should be unified as “GABA levels”.
6. In line 208, the word “reducing” is suggested to be replaced by “inhibiting” or other appropriate words.

Reviewer #2 (Remarks to the Author):

This is an interesting and timely study. The authors conducted comprehensive integrative analyses to identify risk genes for schizophrenia and they successfully prioritized PCCB as a promising candidate. They further performed serial functional experiments to investigate the potential role of PCCB in schizophrenia. Overall, this is a well-designed study and the logic is clear. More importantly, the organoids experiments provided important insights into schizophrenia pathogenesis.

Major comments:

1. In this study, the phenotypes (including ATP, ROS, etc.) of PCCB knockdown hFOs were tested in only one cell line. Using another cell line-derived hFOs to replicate these results could make these findings robust.
2. The authors showed that PCCB knockdown decreased the GABA levels in hFOs. ELISA should be used to validate this key finding. The authors also need to generate hFOs from another iPS cell line to validate this result.
3. As shown in Figure 4G, succinate is a metabolite that connects the TCA cycle and GABA shunt. In addition to succinyl-CoA, the authors need to detect whether succinate is downregulated in PCCB knockdown hFOs using ELISA.
4. Figure 4G is suggested to be re-drawn, since the metabolomic path of PCCB is intersected with the TCA cycle.

Minor comments :

1. Typo errors. In line 46, "deceased" should be "decreased".
2. In lines 91 and 452, "is" should be replaced to "was".
3. In line 160, "PCCB may affect the GABAergic system" should be "PCCB knockdown may affect the GABAergic system".
4. In the entire manuscript, "PCCB-induced DEGs" should be "PCCB knockdown-induced DEGs".
5. In Figure 4F, the label "No" should be replaced by "Control" or "NC".
6. In Fig. S1, "Day60" should be "Day 60".
7. The histogram figures should be presented in a consistent manner, for example, using the scatter dot to represent the biological or technical replicates.
8. In lines 206 and 263, the concentrations of α -KG (10 μ g/ml) and propanoic acid (3.5 μ M) should be marked by reference.
9. Whether PCCB expression was changed in SCZ cases compared with controls ? A quick look in SZDB (PMID: 27451428) will help to answer this pivotal issue.

Reviewer #3 (Remarks to the Author):

In this manuscript, Zhang and colleagues present an analyses pointing to a causal role for PCCB gene in the etiology of schizophrenia. The authors use multiple lines of human genomic evidence to prioritize PCCB for their studies. They confirm gene regulatory roles of SCZ- associated SNPs and eQTLs in the PCCB gene regulate gene expression in a manner similar to the SCZ postmortem brain including regulation of interneuron (GABAergic)-related transcripts in a cerebral organoid model and use multielectrode electrodes to confirm GABA-related e-phys property changes. The paper is well written and I enjoyed reading this manuscript. The authors are experts in the field and the choice of PCCB as a candidate gene is well justified and sound. One concern that diminishes my initial enthusiasm is the lack of convergence between the transcriptomic profiles of the two independent hFOs experiments.

1. I am wondering about the cell type composition of the organoids- how many cells are on a lineage to becoming putative GABAergic neurons? Can the authors stain for GABA markers such as GAD2 or GABAergic developmental markers such as LHX6, ASCL1 or DLX2? Given the prominent role the findings of the paper find for GABA involvement it would be useful to know this. And if its too early in the development of the organoid to see expression of putative GABA markers, what is the rationale for not allowing the organoids to develop further?

2. One concern is the relative lack of overlap between the two hFO transcriptomic profiles (only 1079 of 2326 DEGs replicated, <50%). Are the authors performing technical replicates for these studies or only 2 biological replicates? Can the authors address whether any technical or biological variations between the two donors could be accounting for this? Are there any gene sets in the non-overlapping set that may give clue as to differences in the biology between the two donors?

3. Secondary confirmation (small molecule FISH or digital PCR) of the some of the DEGs would alleviate my concern about the lack of overlap and I believe would strengthen the GABA findings.

Responses to Reviewers' Comments

Dear editor and reviewers:

Thank you very much for giving us the opportunity to revise our manuscript titled "Human forebrain organoids-based multi-omics analyses reveal *PCCB*'s regulation on GABAergic system contributing to schizophrenia". According to your valuable comments, we have made substantial revisions in the manuscript. We believe that the quality of this paper is greatly improved. Here we submit the revised manuscript and our responses to the reviewers' comments for you to consider.

Reviewer #1 (Remarks to the Author):

Major comments:

1. In this study, the authors prioritized six SCZ-associated SNPs that may affect *PCCB* expression. Only three of these six SNPs were chosen for dual luciferase reporter assay. The remaining three SNPs could also be used for the dual luciferase reporter assay.

A: Thank you very much for the reviewer's valuable suggestion. As suggested, the remaining three SNPs (s6791142, rs7616204, and rs570621) have also been used for the dual luciferase reporter assay. The updated results have shown in Figure 1 and the main text (Page 4, lines 109-120).

2. In lines 184-186, the authors should indicate which figure or table was used to show the metabolomic results "Notably, GABA levels were decreased in both *PCCB*- G1 (decreased by 59%, $P = 3.42E-03$) and *PCCB*-G2 hFOs (decreased by 25%, $P = 8.62E-185$) when compared with control hFOs". On the other hand, ELISA should be used to validate the decreased GABA levels in *PCCB* knockdown hFOs. hFOs derived from another iPS cell line are also needed to validate these results.

A: We thank the reviewer's suggestion. As suggested, we have used Figure 3c and Table S3 to indicate the metabolomic results.

We also used ELISA to validate the decreased GABA levels in *PCCB* knockdown U2F hFOs as shown in Figure 4g and the main text (Page 8, lines 215-221). The ELISA results

were also validated in U1M hFOs generated from another iPS cell line (U1M), which are shown in Figure S4i and the main text (Page 8, lines 226-229).

3. In this study, the authors performed transcriptomic and metabolomic analysis after PCCB knockdown in hFOs. The correlations between the differentially expressed genes and metabolites should be analyzed.

A: Per the reviewer's suggestion, we now have performed the correlation analysis between the differentially expressed genes and metabolites. Results of the correlation analysis is shown in Figure 3.

4. As shown in the line 181, 14 differentially expressed metabolites were shared between PCCB-G1 and PCCB-G2 hFOs. The authors should indicate clearly how many metabolites were consistently altered in both PCCB-G1 and PCCB-G2 hFOs.

A: In the revised manuscript, we have written as "A total of 11 differential metabolites were shared in both groups with 9 metabolites showing consistent alteration directions (Figure 3c)". We also indicated the name of 9 shared metabolites in Figure 3c.

Minor comments :

1. There is a typo error in line 46, "deceased" should be corrected to "decreased".

A: We are sorry for the typo. Now we have carefully checked and corrected all typos in the manuscript.

2. In lines 42-45, the sentence "PCCB knockdown in human forebrain organoids (hFOs) followed by RNA-seq revealed dysregulation of genes enriched with multiple neuronal functions including gamma-aminobutyric acid (GABA)-ergic synapse, as well as genes dysregulated in postmortem brains of SCZ patients or in cerebral organoids derived from SCZ patients" should be re-written for more clear understanding.

A: We are sorry for the ambiguous writing. To make it clear, we have shortened the sentence as "PCCB knockdown in human forebrain organoids (hFOs) followed by RNA sequencing

analysis revealed dysregulation of genes enriched with multiple neuronal functions including gamma-aminobutyric acid (GABA)-ergic synapse”.

3. In line 110, there should be an abbreviation for hNPCs.

A: As suggested, we have added full name “human neural progenitor cells” for hNPCs (Page 4, lines 109-110).

4. In Figure 2A, there are no scale bars in two figures of hFOs.

A: Now we have added scale bars in Figure 2A.

5. In the manuscript, “GABA level” should be unified as “GABA levels”.

A: As suggested, we have corrected “GABA level” to “GABA levels”.

6. In line 208, the word “reducing” is suggested to be replaced by “inhibiting” or other appropriate words.

A: Per the reviewer’s suggestion, we have replaced the word “reducing” by “inhibiting”.

Reviewer #2 (Remarks to the Author):

This is an interesting and timely study. The authors conducted comprehensive integrative analyses to identify risk genes for schizophrenia and they successfully prioritized PCCB as a promising candidate. They further performed serial functional experiments to investigate the potential role of PCCB in schizophrenia. Overall, this is a well-designed study and the logic is clear. More importantly, the organoids experiments provided important insights into schizophrenia pathogenesis.

Major comments:

1. In this study, the phenotypes (including ATP, ROS, etc.) of PCCB knockdown hFOs were tested in only one cell line. Using another cell line-derived hFOs to replicate these results could make these findings robust.

A: We sincerely thank the reviewer's positive comments on our study. The reviewer's suggestion is helpful to improve the quality of this paper.

As suggested, we generated hFOs using another iPS cell line (U1M) and validated the results of ELISA and mitochondrial functional assay. Please see the updated results in the main text, Page 8, lines 222-231 and Page 9, 241-242.

2. The authors showed that PCCB knockdown decreased the GABA levels in hFOs. ELISA should be used to validate this key finding. The authors also need to generate hFOs from another iPS cell line to validate this result.

A: Per the reviewer's suggestion, we used ELISA to validate the decreased GABA levels in PCCB knockdown U2F hFOs. The ELISA results are shown in Figure 4g (Page 8, lines 218-221). We also validated the ELISA results in U1M hFOs generated from another iPS cell line (U1M), which is shown in Figure S4i (Page 8, lines 222-231).

3. As shown in Figure 4G, succinate is a metabolite that connects the TCA cycle and GABA shunt. In addition to succinyl-CoA, the authors need to detect whether succinate is downregulated in PCCB knockdown hFOs using ELISA.

A: According to the reviewer's suggestion, we performed ELISA and found that PCCB knockdown led to decreased succinate levels in both U1M and U2F hFOs. Please see the updated results in Figure 4, Figure s4, and the main text (Pages 7-8, lines 210-231).

4. Figure 4G is suggested to be re-drawn, since the metabolomic path of PCCB is intersected with the TCA cycle.

A: We have redrawn Figure 4G. In the revised manuscript, Figure 4G has been updated as Figure 4h.

Minor comments :

1. Typo errors. In line 46, "deceased" should be "decreased".

A: We are sorry for the typo. All typos in the manuscript have been corrected.

2. In lines 91 and 452, “is” should be replaced to “was”.

A: Thanks for the correction. The word “is” has been replaced to “was”.

3. In line 160, “PCCB may affect the GABAergic system” should be “PCCB knockdown may affect the GABAergic system”.

A: As suggested, “PCCB may affect the GABAergic system” has been corrected to “PCCB knockdown may affect the GABAergic system”.

4. In the entire manuscript, “PCCB-induced DEGs” should be “PCCB knockdown-induced DEGs”.

A: In the revised manuscript, “ PCCB-induced DEGs” has been corrected to “PCCB knockdown-induced DEGs”.

5. In Figure 4F, the label "No" should be replaced by "Control" or “NC”.

A: We are sorry for the errors. The errors have been corrected.

6. In Fig. S1, “Day60” should be “Day 60”.

A: The typo has been corrected as suggested.

7. The histogram figures should be presented in a consistent manner, for example, using the scatter dot to represent the biological or technical replicates.

A: In the revised manuscript, we used scatter dot plots to present all the statistical results.

8. In lines 206 and 263, the concentrations of α -KG (10 μ g/ml) and propanoic acid (3.5 μ M) should be marked by reference.

A: In the revision, reference 45 and 52 have been added. To be noted, the concentration of propanoic acid has been corrected to 3.2 μ M due to the typo error in the manuscript.

9. Whether PCCB expression was changed in SCZ cases compared with controls? A quick look in SZDB (PMID: 27451428) will help to answer this pivotal issue.

A: Thanks for the reviewer's suggestion. We have checked PCCB expression in the SZDB database and cited the reference. We updated the results in the main text as "*PCCB* was also found to be nominally downregulated in postmortem SCZ brains ($P = 0.01$, FDR = 0.14) by checking the SZDB database²⁰ which integrated transcriptome data from the CommonMind consortium²¹." (Pages 3-4, lines 89-91).

Reviewer #3 (Remarks to the Author):

In this manuscript, Zhang and colleagues present an analyses pointing to a causal role for PCCB gene in the etiology of schizophrenia. The authors use multiple lines of human genomic evidence to prioritize PCCB for their studies. They confirm gene regulatory roles of SCZ- associated SNPs and eQTLs in the PCCB gene regulate gene expression in a manner similar to the SCZ postmortem brain including regulation of interneuron (GABAergic)-related transcripts in a cerebral organoid model and use multielectrode electrodes to confirm GABA-related e-phys property changes. The paper is well written and I enjoyed reading this manuscript. The authors are experts in the field and the choice of PCCB as a candidate gene is well justified and sound. One concern that diminishes my initial enthusiasm is the lack of convergence between the transcriptomic profiles of the two independent hFOs experiments.

1. I am wondering about the cell type composition of the organoids- how many cells are on a lineage to becoming putative GABAergic neurons? Can the authors stain for GABA markers such as GAD2 or GABAergic developmental markers such as LHX6, ASCL1 or DLX2? Given the prominent role the findings of the paper find for GABA involvement it would be useful to know this. And if its too early in the development of the organoid to see expression of putative GABA markers, what is the rationale for not allowing the organoids to develop further?

A: We sincerely thank the reviewer's positive comments on our study. We agree with the reviewer that it's important to confirm the existence of GABAergic neurons in hFOs (day 60). Through the immunostaining assay, we confirmed the existence of GABAergic neurons that

expressed the putative GABA markers GAD2+ (GAD67+) and DLX2+ in hFOs (Fig. S7). The percentage of GAD2+ area or DLX2+ cells is about 5% in the total cells of hFOs.

Since the proportion of GABAergic neurons is limited in day 60 hFOs, we discussed the limitation in the updated manuscript (Page 11, lines 307-311) as “Second, hFOs (day 60) are immature in this study since the average proportion of GAD67+ or DLX2+ GABAergic neurons only account for about 5% of the total cells in hFOs (Fig. S7). A long-term organoid culture could allow hFOs develop further to generate abundant GABAergic neurons to validate the GABA findings”.

2. One concern is the relative lack of overlap between the two hFO transcriptomic profiles (only 1079 of 2326 DEGs replicated, <50%). Are the authors performing technical replicates for these studies or only 2 biological replicates? Can the authors address whether any technical or biological variations between the two donors could be accounting for this? Are there any gene sets in the non-overlapping set that may give clue as to differences in the biology between the two donors?

A: In our study, two hFO transcriptomic profiles were derived from the same iPS cell line (U2F). However, these two hFO transcriptome were generated from different PCCB knockdown gRNA sequences (PCCB-G1 and PCCB-G2) and performed in different batch. In the RNA-seq experiment, two hFOs were pooled together as a mixed sample, and at least three mixed samples were used for RNA-seq analysis.

Since the two hFO transcriptomic profiles are generated from the same iPS cell-derived hFOs, we think that the difference of gRNA sequence may account for the major variation between these two transcriptomic profiles. Additionally, the sample size used for RNA-seq is limited, using more samples for RNA-seq could make the results robust and may improve the overlapping of DEGs between PCCB-G1 and PCCB-G2 hFOs.

Though only about 50% DEGs were overlapped between the two hFO transcriptomic profiles, key RNA-seq findings were validated in hFOs derived from three independent iPS cell lines. For example, the decreased expression of GABAergic synapse-related genes (GABRA1, GABRA2, GABRB2, and GABRB3) were all decreased in PCCB knockdown hFOs

derived from U2F, U1M, and ACS-1011 iPS cell lines (Figure 2j and Figure S6c-d). The metabolomic profiling and ELISA analysis also confirmed the GABA findings. We believed that the key findings in this study are robust.

3. Secondary confirmation (small molecule FISH or digital PCR) of the some of the DEGs would alleviate my concern about the lack of overlap and I believe would strengthen the GABA findings.

A: We understand the reviewer's concern. GABA findings are key results of this study, and several GABAergic synapse-related DEGs including GABRA1, GABRA2, GABRB2, and GABRB3 were highlighted as hub nodes in PCCB's regulatory network. We performed RT-qPCR analysis to validate expression of GABRA1, GABRA2, GABRB2, and GABRB3 in hFOs generated from three independent iPS cell lines (U1M, U2F, and ACS-1011).

The RT-qPCR analysis showed that expression of GABRA1, GABRA2, GABRB2, and GABRB3 were decreased in PCCB knockdown hFOs derived from all the three iPS cell lines (Figure 2j and Figure S6c-d). These RT-qPCR results were consistent with those observed in the RNA-seq analysis, suggesting the robust GABA findings in our study.

REVIEWERS' COMMENTS

Reviewer #1 (Remarks to the Author):

I have no further comments.

Reviewer #2 (Remarks to the Author):

The authors addressed all of my concerns and comments, I have no further comments.

Reviewer #3 (Remarks to the Author):

The authors have addressed all of my concerns.

Response to Reviewers' Comments

Dear editor and reviewers:

Thank you very much for giving us the opportunity to revise our manuscript (Manuscript ID: NCOMMS-23-11278A) for acceptance. According to the reviewers' comments and editorial requests, we have made corresponding revisions in the manuscript. Here we submit our response to the reviewers' comments for you to check.

Reviewer #1 (Remarks to the Author):

I have no further comments.

A: We sincerely thank the reviewer's help and guidance on the manuscript.

Reviewer #2 (Remarks to the Author):

The authors addressed all of my concerns and comments, I have no further comments.

A: Thanks for the reviewer's valuable comments and suggestions on the manuscript.

Reviewer #3 (Remarks to the Author):

The authors have addressed all of my concerns.

A: We are pleased that we could address the reviewer's concerns. We appreciate the reviewer's comments and guidance on the manuscript.